# Sigma-Delta Quantized Networks

**Peter O'Connor, Max Welling**
QUVA Lab, Informatics Institute
University of Amsterdam
Amsterdam, Netherlands
`{p.e.oconnor,m.welling}@uva.nl`

## ABSTRACT

Deep neural networks can be obscenely wasteful. When processing video, a convolutional network expends a fixed amount of computation for each frame with no regard to the similarity between neighbouring frames. As a result, it ends up repeatedly doing very similar computations. To put an end to such waste, we introduce Sigma-Delta networks. With each new input, each layer in this network sends a discretized form of its *change* in activation to the next layer. Thus the amount of computation that the network does scales with the amount of change in the input and layer activations, rather than the size of the network. We introduce an optimization method for converting any pre-trained deep network into an optimally efficient Sigma-Delta network, and show that our algorithm, if run on the appropriate hardware, could cut at least an order of magnitude from the computational cost of processing video data.

## 1 INTRODUCTION

For most deep-learning architectures, the amount of computation required to process a sample of input data is independent of the contents of that data.

Natural data tends to contain a great deal of spatial and temporal redundancy. Researchers have taken advantage of such redundancy to design encoding schemes, like jpeg and mpeg, which introduce small compromises to image fidelity in exchange for substantial savings in the amount of memory required to store images and videos.

In neuroscience, it seems clear that that some kind of sparse spatio-temporal coding is going on. Koch et al. (2006) estimate that the human retina transmits 8.75Mbps, which is about the same as compressed 1080p video at 30FPS.

Thus it seems natural to think that perhaps we should be doing this in deep learning. In this paper, we propose a neural network where neurons only communicate discretized changes in their activations to one another. The computational cost of running such a network would be proportional to the amount of change in the input. Neurons send signals when the change in their input accumulates past some threshold, at which point they send a discrete "spike" notifying downstream neurons of the change. Such a system has at least two advantages over the conventional way of doing things.

1. When extracting features from temporally redundant data, it is much more efficient to communicate the changes in activation than it is to re-process each frame.

2. When receiving data asynchronously from different sources (e.g. sensors, or nodes in a distributed network) at different rates, it no longer makes sense to have a global network update. We could recompute the network with every new input, reusing the stale inputs from the other sources, but this requires doing a great deal of repeated computation for only small differences in input data. We could keep a history of all inputs and update the network periodically, but then we lose the ability to respond immediately to new inputs. Our approach gets around this ugly tradeoff by allowing for efficient approximate updates of the network given a partial update to the input data. The computational cost of the update is proportional to the effect that the new information has on the network's state.

## 2   RELATED WORK

This work originated in the study of spiking neural networks, but treads into the territory of discretizing neural nets. The most closely related work is that of Zambrano and Bohte (2016). In this work, the authors describe an Adaptive Sigma-Delta modulation method, in which neurons communicate analog signals to one another by means of a "spike-encoding" mechanism, where a temporal signal is encoded into a sequence of weighted spikes and then approximately decoded as a sum of temporally-shifted exponential kernels. The authors create a scheme for being parsimonious with spikes by allowing adaptive scaling of thresholds, at the cost of sending spikes with real values attached to them, rather than the classic "all or nothing" spikes. Their work references a slightly earlier work by Yoon (2016) which reframes common neural models as forms of Asynchronous Sigma-Delta modulation. In a concurrent work, Lee et al. (2016) implement backpropagation in a similar system (but without adaptive threshold scaling), and demonstrate the best-yet performance on MNIST for networks trained with spiking models. This work postdates Diehl et al. (2015), which proposes a scheme for normalizing neuron activations so that a spiking neural network can be optimized for fast classification.

Our model contrasts with all of the above in that it is time-agnostic. Although we refer to sending "temporal differences" between neurons, our neurons have no concept of time - their is no "leak" in neuron potential, and our neurons' behaviour only depends on the order of the inputs. Our work also separates the concepts of nonlinearity and discretization, uses units that communicate differences rather than absolute signal values, and explicitly minimizes an objective function corresponding to computational cost.

Coming from another corner, Courbariaux et al. describe a scheme for binarizing networks with the aim of achieving reductions in the amount of computation and memory required to run neural nets. They introduce a number of tricks for training binarized neural networks - a normally difficult task due to the lack of gradient information. Esser et al. (2016) use a similar binarization scheme to efficiently implement a spiking neural network on the IBM TrueNorth chip. Ardakani et al. (2015) take another approach - to approximate real-valued operations of a neural net with a sequence of stochastic integer operations, and show how these can lead to cheaper computation.

These discretization approaches differ from ours in that they do not aim to take advantage of temporal redundancy in data, but rather aim to find ways of saving computation by learning in a low-precision regime. Ideas from these works could be combined with the ideas presented in this paper.

The idea of sending quantized temporal differences has been applied to make event-based sensors, such as the Dynamic-Vision Sensor (Lichtsteiner et al., 2008), which quantize changes in analog pixel-voltages and send out pixel-change events asynchronously. The model we propose in this paper could be used to efficiently process the outputs of such sensors.

Finally, our previous work, (O'Connor and Welling, 2016) develops a method for doing backpropagation with the same type of time-agnostic spiking neurons we use here. In this paper, we do not aim to train the network from scratch, but instead focus on how we can compute efficiently by sending temporal differences between neurons.

## 3   THE SIGMA-DELTA NETWORK

In this Section, we describe how we start with a traditional deep neural network and apply two modifications - temporal-difference communication and rounding - to create the Sigma-Delta network. To explain the network, we follow the Figure 1 from top to bottom, starting with a standard deep network and progressing to our Sigma-Delta network. Here, we will think of the forward pass of a neural network as composition of subfunctions: $f(x) = (f_L \circ ... \circ f_2 \circ f_1)(x)$.

### 3.1   TEMPORAL DIFFERENCE NETWORK

We now define "temporal difference" ($\Delta_T$) and "temporal integration" ($\Sigma_T$) modules as follows:

| **Algorithm 1** Temporal Difference ($\Delta_T$): | **Algorithm 2** Temporal Integration ($\Sigma_T$): |
|---|---|
| 1: **Internal:** $\vec{x}_{last} \in \mathbb{R}^d \leftarrow \vec{0}$ | 1: **Internal:** $\vec{y} \in \mathbb{R}^d \leftarrow \vec{0}$ |
| 2: **Input:** $\vec{x} \in \mathbb{R}^d$ | 2: **Input:** $\vec{x} \in \mathbb{R}^d$ |
| 3: $\vec{y} \leftarrow \vec{x} - \vec{x}_{last}$ | 3: $\vec{y} \leftarrow \vec{y} + \vec{x}$ |
| 4: $\vec{x}_{last} \leftarrow \vec{x}$ | 4: **Return:** $\vec{y} \in \mathbb{R}^d$ |
| 5: **Return:** $\vec{y} \in \mathbb{R}^d$ | |

So that when presented with a sequence of inputs $x_1, ...x_t$, $\Delta_T(x_t) = x_t - x_{t-1}|_{x_0=0}$, and $\Sigma_T(x_t) = \sum_{\tau=1}^{t} x_\tau$. It should be noted that when we refer to "temporal differences", we refer not to the change in the signal over time, but in the change between two inputs presented sequentially. The output of our network only depends on the value and order of inputs, not on the temporal spacing between them. This distinction only matters when dealing with asynchronous inputs such as the Dynamic Vision Sensor, (Lichtsteiner et al., 2008), which are not considered in this paper.

Since $\Sigma_T(\Delta_T(x_t)) = x_t$, we can insert $\Sigma_T \circ \Delta_T$ pairs into the network without affecting the function. So we can re-express our network function as: $f(x) = (f_L \circ \Sigma_T \circ \Delta_T \circ ... \circ f_2 \circ \Sigma_T \circ \Delta_T \circ f_1 \circ \Sigma_T \circ \Delta_T)(x)$.

Now suppose our network consists of alternating linear functions $w(x)$, and nonlinear functions $h(x)$, so that $f(x) = (h_L \circ w_L ... \circ h_2 \circ w_2 \circ h_1 \circ w_1)(x)$. As before, we can harmlessly insert our $\Sigma_T \circ \Delta_T$ pairs into the network. But this time, note that for a linear function $w(x)$, the operations $(\Sigma_T, w, \Delta_T)$ all commute with one another. That is:

$$\Delta_T(w(\Sigma_T(x))) = w(\Delta_T(\Sigma_T(x))) = w(x) \tag{1}$$

Therefore we can replace all instances of $\Delta_T \circ w \circ \Sigma_T$ with $w$, yielding $f(x) = (h_L \circ \Sigma_T \circ w_L \circ ... \circ \Delta_T \circ h_2 \circ \Sigma_T \circ w_2 \circ \Delta_T \circ h_1 \circ \Sigma_T \circ w_1 \circ \Delta_T)(x)$, which corresponds to the network shown in Figure 1 B. For now this is completely pointless, since we do not change the network function at all, but it will come in handy in the next section, where we discretize the output of the $\Delta_T$ modules.

## 3.2 Discretizating the Deltas

When dealing with data that is naturally spatiotemporally redundant, like most video, we expect the output of the $\Delta_T$ modules to be a vector with mostly low values, with some peaks corresponding to temporal transitions at certain input positions. We expect the data to have this property not only at the input layer, but even more so at higher layers, which encode higher level features (edges, object parts, class labels), which we would expect to vary more slowly over time than pixel values. If we discretize this "peaky" vector, we end up with a sparse vector of integers, which can then be used to cheaply communicate the approximate change in state of a layer to its downstream layer(s).

A sensible approach is to apply rounding before the temporal-difference operation - i.e. round the activation values and then send the temporal differences of these rounded values. It is then easy to show that the network's function will remain identical to that of the rounding network:

$$\Sigma_T(w(\Delta_T(round(x)))) = w(\Sigma_T(\Delta_T(round(x)))) = w(round(x)) \tag{2}$$

It's worth noting that this is equivalent to doing discrete-time Sigma-Delta modulation to quantize the temporal differences - this connection is explained in Appendix A.

It follows from this result that our Sigma-Delta network depicted in Figure 1 D computes an identical function to that of the rounding network in Figure 1 C. In other words, the output $y_t$ of the Sigma-Delta network is solely dependent on the parameters of the network and the current input $x_t$, and not on any of the previous inputs $x_1..x_{t-1}$. The amount of computation required for the update, however, depends on $x_{t-1}$. Specifically, if $x_t$ is similar to $x_{t-1}$, the Sigma-Delta network should require less computation to perform an update than the Rounding Network.

### 3.3 SPARSE DOT PRODUCT

Most of the computation in Deep Neural networks is consumed doing matrix multiplications and convolutions. The architecture we propose saves computation by translating the input to these operations into an integer array with a small L1 norm.

With sparse, low-magnitude integer input, we can compute the vector-matrix dot product efficiently by decomposing it into a sequence of vector additions. We can see this by decomposing the vector $\vec{x} \in \mathbb{I}^{d_{in}}$ into a set of indices $\langle (i_n, s_n) : i \in [1..len(\vec{x})], s \in \pm 1, n = [1..N] \rangle$, such that: $\vec{x} = \sum_{n=1}^{N} s_n \vec{e}_{i_n}$, where $e_{i_n}$ is a one-hot vector with element $i_n$ hot, and $N = |\vec{x}|_{L1}$ is the total L1 magnitude of the vector. We can then compute the dot-product as a series of additions, as shown in Equation 3.

$$
\begin{aligned}
u = \vec{x} \cdot W &: W \in \mathbb{R}^{d_{in} \times d_{out}} \\
&= \left( \sum_{n=1}^{N} s_n \vec{e}_{i_n} \right) \cdot W = \sum_{n=1}^{N} \vec{s}_n e_{i_n} \cdot W = \sum_{n=1}^{N} s_n \cdot W_{i_n,}.
\end{aligned}
\tag{3}
$$

Computing the dot product this way takes $N \cdot d_{out}$ additions. A normal dense dot-product, by comparison, takes $d_{in} \cdot d_{out}$ multiplications and $(d_{in} - 1) \cdot d_{out}$ additions.

This is where the energy savings come in. Horowitz (2014) estimates that on current 45nm silicon process, a 32-bit floating-point multiplication costs 3.7pJ, vs 0.9pJ for floating-point addition. With integer or fixed-point arithmetic, the difference is even more pronounced, with 3.1pJ for multiplication vs 0.1pJ for addition. This of course ignores the larger cost of processing instructions and moving memory, but gives us an idea of how these operations might compare on optimized hardware. So provided we can approximate the forward pass of a network to a satisfactory degree of precision without doing many more operations than the original network, we can potentially compute much more efficiently.

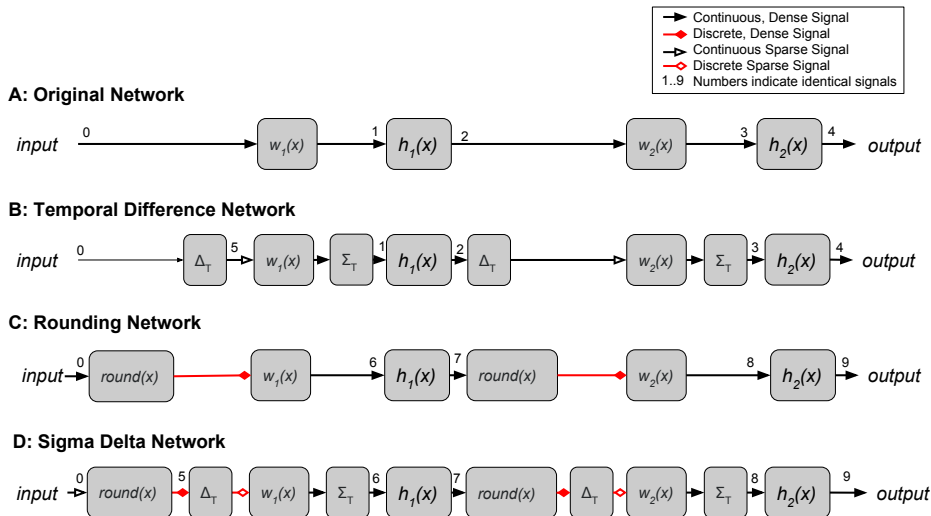

Figure 1: **A**: An ordinary deep network, which consists of an alternating sequence of linear operations $w_i(x)$, and nonlinear transforms $h_i(x)$. **B**: The Temporal-Difference Network, described in Section 3.1, computes the exact same function as network A, but communicates differences in activation between layers. **C**: An approximation of network A where activations are rounded before being sent to the next layer. **D**: The Sigma-Delta Network combines the modifications of B and C. Functionally, it is identical to the Rounding Network, but it can compute forward passes more cheaply when input data is temporally redundant.

### 3.4 PUTTING IT ALL TOGETHER

Figure 1 visually summarizes the four types of network we have described. Inserting the temporal sum and difference modules discussed in Section 3.1 leads to the Temporal Difference Network, which

is functionally identical to the Original Network. Discretizing the output of the temporal difference modules, as discussed in Section 3.2, leads to the Sigma-Delta network. The Sigma-Delta Network is functionally equivalent to the Rounding network, except that it requires less computation per forward pass if it is fed with temporally redundant data.

# 4  OPTIMIZING AN EXISTING NETWORK

In this work, we do not aim to train a quantized networks from scratch, as we did in O'Connor and Welling (2016). Rather, we will take existing pretrained networks and optimize them as Sigma-Delta networks. In in our situation, we have two competing objectives: Error (with respect to a non-quantized forward pass), and Computation: the number of additions performed in a forward pass.

## 4.1  RESCALING OUR NEURONS

We can control the trade-off between these objectives by changing the scale of our discretization. We can thus extend our rounding function by adding a scale $k \in \mathbb{R}^+$:

$$round(\vec{x}, k) \equiv round(\vec{x} \cdot k)/k \tag{4}$$

This scale can either be layerwise or unitwise (in which case we have a vector of scales per layer). Higher $k$ values will lead to higher precision, but also more computation, for the reason mentioned in Section 3.2. Note that the final division-by-k is equivalent to scaling the following weight matrix by $\frac{1}{k}$,. So in practice, our network functions become:

$$f_{round}(x) = \left( h_L \circ \frac{w_L}{k_L} \circ round \circ \cdot k_L \circ ... \circ h_1 \circ \frac{w_1}{k_1} \circ round \circ \cdot k_1 \right)(x) \tag{5}$$

$$f_{\Sigma\Delta}(x) = \left( h_L \circ \Sigma_T \circ \frac{w_L}{k_L} \circ round \circ \cdot k_L \circ \Delta_T \circ ... \circ h_1 \circ \Sigma_T \circ \frac{w_1}{k_1} \circ round \circ \cdot k_1 \circ \Delta_T \right)(x) \tag{6}$$

For the Rounding Network and the Sigma-Delta Network, respectively. By adjusting these scales $k_l$, we can affect the tradeoff between computation and error. Note that if we use ReLU activation functions, parameters $k_l$ can simply be baked into the parameters of the network (see Appendix C.)

## 4.2  THE ART OF COMPROMISE

In this section, we aim to find the optimal trade-offs between Error and Computation for the Rounding Network (Network C in Figure 1). We define our loss as follows:

$$\mathcal{L}_{error} = \mathcal{D}(f_{round}(x), f_{true}(x)) \tag{7}$$

$$\mathcal{L}_{comp} = \sum_{l=1}^{L-1} |s_l|_{L1} d_{l+1} \tag{8}$$

$$\mathcal{L}_{total} = \mathcal{L}_{error} + \lambda \mathcal{L}_{comp} \tag{9}$$

Where $\mathcal{D}(a, b)$ is some scalar distance function (We use KL-divergence for softmax output layers and L2-norm otherwise), $f_{round}(x)$ is the output of the Rounding Network, $f_{true}(x)$ is the output of the Original Network. $\mathcal{L}_{comp}$ is the computational loss, defined as the total number of additions required in a forward pass. Each layer performs $|s_l|_{L1} d_{l+1}$ additions, where $s_l$ is the discrete output of the $l$'th layer, $d_{l+1}$ is the dimensionality of the $(l+1)$'th layer. Finally $\lambda$ is the tradeoff parameter balancing the importance of the two losses.

We aim to use this loss function to optimize our layer-scales, $k_l$ to find an optimal tradeoff between accuracy and computation, given the tradeoff parameter $\lambda$.

### 4.3 DIFFERENTIATING THE UNDIFFERENTIABLE

We run into an obvious problem: $y = round(k \cdot x)$ is not differentiable with respect to our scale, $k$ or our input, $x$. We get around this by using a similar method to Courbariaux et al., who in turn borrowed it from a lecture by Hinton (2012). That is, on the backward pass, when computing the gradient with respect to the error $\frac{\partial \mathcal{L}_{error}}{\partial k_l}$, we simply pass the gradient through the rounding function in layers $[l+1, .., .L]$, i.e. we say $\frac{\partial}{\partial x} round(x) \approx 1$.

When computing the gradient with respect to the computational cost, $\frac{\partial \mathcal{L}_{comp}}{\partial k_l}$, we again just pass the gradient through all rounding operations in the backward pass for layers $[l+1, .., .L]$. We found instabilities in training when using the computational loss of higher layers: $\mathcal{L}_{comp,l'} : l' \in [l+1, ..., L]$, to update the scale of layer $l$. Since we don't expect this term to have much effect anyway, we choose to only use the gradient of the computational cost in layer $l$ when updating scale $k_l$, i.e., we approximate: $\frac{\partial \mathcal{L}_{comp}}{\partial k_l} \approx \frac{\partial \mathcal{L}_{comp,l}}{\partial k_l}$.

Our scale parameters also must remain in the positive range, and stay well away from zero, where they can cause instability due to the division-by-k (see Equation 5). To handle this, we parametrize our scales in log-space, as $\kappa_l = log(k_l)$. Our scale-parameter update rule becomes:

$$\Delta \kappa_l = -\eta \left( \frac{\partial \mathcal{L}_{error}}{\partial \kappa_l} \bigg|_{pass:[l+1..L]} + \lambda \frac{\partial}{\partial \kappa_l} |\vec{s}_l|_{L1} d_{l+1} \bigg|_{pass:l} \right) \quad (10)$$

Where $\vec{s}_l$ is the rounded signal from layer $l$, $d_{l+1}$ is the "fan-out" (equivalent to the dimension of layer $l+1$ in a fully-connected network), and $pass : [l+1..L]$ indicates that, on the backward pass, we simply pass the gradient through the rounding functions on layers $[l+1..L]$.

## 5 EXPERIMENTS

### 5.1 TOY PROBLEM: A RANDOM NETWORK

We start with a very simple toy problem to verify our method. We initialize a 2-layer (100-100-100) ReLU network with random weights using the initialization scheme proposed in Glorot and Bengio (2010), then scaled the weights by $\left(\frac{1}{2}, 8, \frac{1}{4}\right)$. The weight-rescaling does not affect the function of the network but makes it very ill-adapted for discretization (the first layer will be represented too coarsely, causing error; the second too finely, causing wasted computation). We create random input data, and use it to optimize the layer scales according to Equation 10. We verify, by comparing to a large collection of randomly drawn rescalings, that by tuning lambda we land on different places of the Pareto frontier balancing error and computation. Figure 2 shows that this is indeed the case. In this experiment, error and computation are evaluated just on the Rounding network - we test the Sigma-Delta network in the next experiment, which includes temporal data.

### 5.2 TEMPORAL-MNIST

In order to evaluate our network's ability to save computation on temporal data, we create a dataset that we call "Temporal-MNIST". This is just a reshuffling of the standard MNIST dataset so that similar frames tend to be nearby, giving the impression of a temporal sequence (see Appendix D for details). The columns of Figure 3 show eight snippets from the Temporal-MNIST dataset.

We started our experiment with a conventional ReLU network with layer sizes [784-200-200-10] pre-trained on MNIST to a test-accuracy of 97.9%. We then apply the same scale-optimization procedure for the Rounding Network used in the previous experiment to find the optimal rescalings under a range of values for $\lambda$. This time, we test the learned scale parameters on both the Rounding Network and the Sigma-Delta network. We do not attempt to directly optimize the scales with respect to the amount of computation in the Sigma-Delta network - we assume that the result should be similar to that for the rounding network, but verifying this is the topic of future work.

The results of this experiment can be seen in Figure 4. We see that our discretized networks (Rounding and Sigma-Delta) converge to the error of the original network with fewer computations than are required for a forward pass of the original neural network. Note that the errors of the rounding and

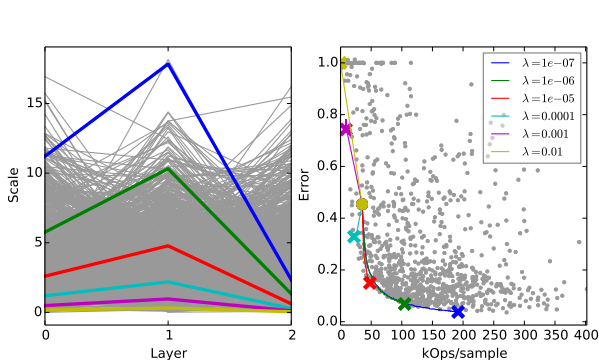

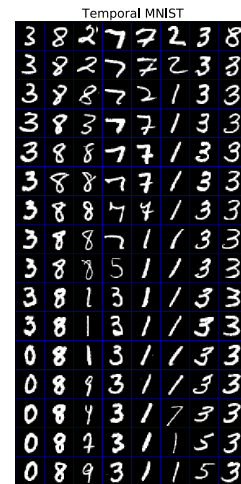

Figure 2: The Results of the "Random Network" experiment described in Section 5.1. **Left**: A plot of the layerwise scales. Grey lines show randomly sampled scales, and coloured lines show optimal scales for different values of $\lambda$. **Right**: Gray dots show the error-scale tradeoffs of network instantiations using the (gray) randomly sampled rescalings on the left. Coloured lines show the optimization trajectory under different values of $\lambda$, starting with the initial state (•), and ending with ×.

Figure 3: Some samples from the Temporal-MNIST dataset. Each column shows a snippet of adjacent frames.

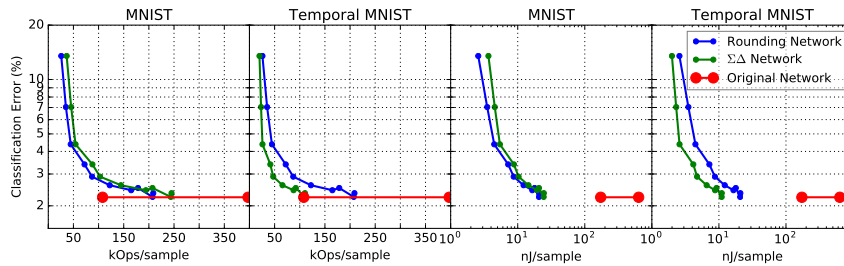

Figure 4: A visualization of our error-computation tradeoff curve for MNIST and our Temporal-mnist dataset. **Plot 1**: Each point on the line for the Rounding (blue) and Sigma-Delta (green) network correspond to the performance of the network for a different value of the error-computation tradeoff parameter $\lambda$, ranging from $\lambda = 10^{-10}$ (in the high-computation, low-error regime) to $\lambda = 10^{-5}$ (in the low-computation, high-error regime). The red line indicates the performance of the original, non-discretized network. The red dot on the right indicates the number of flops required for a full forward pass when doing dense multiplication, and the dot on the left indicates the number of flops when factoring in layer sparsity. Note that for the Rounding and Sigma-Delta networks, we count the number of additions, and for the original network we count the numbers of multiplications and additions (as per Section 3.3). **Plot 2**: The same, but on the Temporal-MNIST dataset. We see that the Sigma-Delta network uses less computation thanks to the temporal redundancy in the data. **Plots 3 and 4**: Half of the original network's Ops were multiplies, which are more computational costly than the additions of the Rounding and Sigma-Delta networks. In these plots the x-axis is rescaled according to the energy use calculations of Horowitz (2014), assuming the weights and parameters of the network are implemented with 32-bit integer arithmetic. Numbers are in Appendix E.

Sigma-Delta networks are identical. This is a consequence of their equivalence, described in Section 3.2. Note also that the errors for all networks are identical between the MNIST and Temporal-MNIST datasets, since for all networks, the prediction function is independent of the order in which inputs are processed. We see that as expected, our Sigma-Delta network does fewer computations than the rounding network on the Temporal-MNIST dataset for the same error, because the update-mechanism of this network takes advantage of the temporal redundancy in the data.

### 5.3 A Deep Convolutional Network on Video

Our final experiment is a preliminary exploration into how Sigma Delta networks could perform on natural video data. We start with "VGG 19" - a 19 layer convolutional network, trained to recognise the 1000 ImageNet categories. The network was trained and made public by Simonyan and Zisserman (2014). We take selected videos from the ILSVRC 2015 dataset (Russakovsky et al., 2015), and apply the rescaling method from Section 4.1 to adjust the scales on a per-layer basis. We initially had some difficulty in optimizing the scale parameters of network to a stable point. The network would either fail to reduce computation when it could afford to, or reduce it to the point where the network's function was so corrupted that error gradients would be meaningless, causing computation loss to win out and activations to drop to zero. A simple solution was to replace the rounding operation in training with addition of uniform random noise $\epsilon \sim U(-\frac{1}{2}, \frac{1}{2})$. This seemed to prevent the network from pushing itself into a regime where all activations become zero. More work is need to understand why the addition of noise is necessary here. Figure 5 shows some preliminary results, which indicate that for video data we can get about 4-10x savings in the amount of computation required, in exchange for a modest loss in computational accuracy.

## 6 Discussion

We have introduced Sigma-Delta Networks, which give us a new way compute the forward pass of a deep neural network. In Sigma-Delta Networks, neurons communicate not by telling other neurons about their current level of activation, but about their change in activation. By discretizing these changes, we end up with very sparse communication between layers. The more similar two consecutive inputs $(x_t, x_{t+1})$ are, the less computation is required to update the network. We show that, while the Sigma-Delta Network's internal state at time-step $t$ depends on past inputs $x_1..x_{t-1}$, the output $y_t$ only depends on the current input $x_t$. We show that there is a tradeoff between the accuracy of this network (with respect to the function of a traditional deep net with the same parameters), and the amount of computation required. Finally, we propose a method to jointly optimize error and computation, given a tradeoff parameter $\lambda$ that indicates how much accuracy we are willing to sacrifice in exchange for fewer computations. We demonstrate that this method substantially reduces the number of computations required to run a deep network on natural, temporally redundant data. However, we observe in our final experiment (Figure 5, bottom) that our assumption that higher-level features would be more temporally stable - and thus require less computation in our Sigma-Delta net - was not true. We suspect that if we were to train the network from scratch on temporal data, we may learn more temporally stable "slow" features, but this is a topic of future work.

A huge amount of data (eg. video, audio) comes in the form of temporal sequences, and there is an increasingly obvious need to be able to process this data efficiently. There is much to be gained by only doing processing when necessary, based on the contents of the data, and we provide one method for doing that. Further work is needed to determine whether this method would be of use on modern computing hardware, namely GPUs. The problem is that these devices are designed for large, fixed-size array operations, and tend not to be good at taking advantage of sparsity in the data, which requires many random memory accesses to parameters. Fortunately, other devices such as the the IBM TrueNorth (Cassidy et al., 2013) are being designed which keep memory close to processing, and such handle sparse data (and random memory access) much more efficiently.

This work opens up an interesting door. In asynchronous, distributed neural networks, a node may receive input from many different nodes asynchronously. Recomputing the function of the network every time a new input signals arrives may be prohibitively expensive. Our scheme deals with this by making the computational cost of an update proportional to the amount of change in the input. The next obvious step is to extend this approach to communicating changes in gradients, which may be helpful in setting up distributed, asynchronous schemes for training neural networks.

Code for our experiments can be found at: `https://github.com/petered/sigma-delta/`

### Acknowledgments

This work was supported by Qualcomm, who we'd also like to thank for discussing their past work in the field with us. We'd also like to thank fellow lab members, especially Changyong Oh and Matthias Reisser, for fruitful discussions contributing to this work.

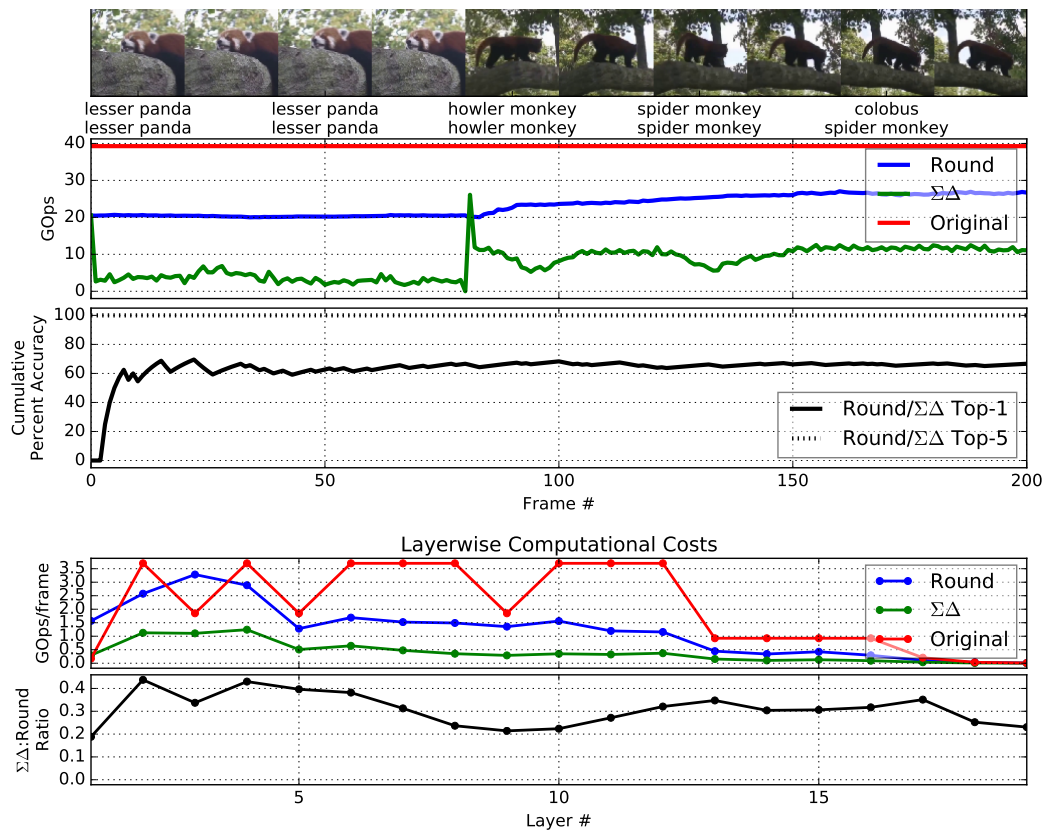

Figure 5: A comparison of the Original VGG Net with the Rounding and Sigma-Delta Networks using the same parameters, after scale-optimization. **Top**: Frames taken from two videos from the ILSVRC2015 dataset. The two videos, with 201 frames in total, were spliced together. The first has a static background, and the second has more motion. Below every second image is the label generated for that image by VGGnet and the Sigma-Delta network (which is functionally equivalent to the Rounding Network, though numerical errors can lead to small changes, not shown here). Scale parameters were trained on separate videos. **Second Plot**: A comparison of the computational cost per-frame. The original VGG network has a fixed cost. The Sigma-Delta network has a cost that varies with the amount of action in the video. The spike in computation occurs at the point where the videos are spliced together. We can see that the Sigma-Delta network does more computation for the second video, in which there is more movement. During the first video it performs about 11 times less computation than the Original Network, during the second, about 4 times less. The difference would be more pronounced if we were to count energy use, as we did in Figure 4. **Third Plot**: A plot of the cumulative mean error (over frames) of the Sigma-Delta/Rounding networks, as compared to the Original VGGnet. Most of the time, it gets the same result (Top-1) out of 1000 possible categories. On almost every frame, the guess of the Sigma-Delta network is one of the top-5 guesses of the original VGGNet. **Fourth Plot**: A breakdown of how much of the computational cost of each network comes from each layer. **Fifth Plot**: The layer-wise ratio of the computational cost of the Sigma-Delta net to the rounding net. We had expected (and hoped) this ratio to become very low in the upper layers, as the high-level features should not change much between frames. However this was not the case (as the ratio remains between 0.2 and 0.4 across all layers). It appears therefore that our assumption - that higher level features would be more temporally stable - is untrue. Appendix F shows that this is a property of the pretrained network, not our quantization scheme.

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

## A  DELTA-HERDING PROOF

| **Algorithm 3** Herding | **Algorithm 4** Delta-Herding |
|---|---|
| 1: **Internal:** $\vec{\phi} \in \mathbb{R}^d \leftarrow \vec{0}$ | 1: **Internal:** $\vec{s}_{last} \in \mathbb{I}^d \leftarrow \vec{0}$ |
| 2: **Input:** $\vec{x_t} \in \mathbb{R}^d$ | 2: **Input:** $\vec{x_t} \in \mathbb{R}^d$ |
| 3: $\vec{\phi} \leftarrow \vec{\phi} + \vec{x_t}$ | 3: $\vec{s} \leftarrow round(x_t)$ |
| 4: $\vec{s} \leftarrow round(\vec{\phi})$ | 4: $\Delta\vec{s} \leftarrow \vec{s} - \vec{s}_{last}$ |
| 5: $\vec{\phi} \leftarrow \vec{\phi} - \vec{s}$ | 5: $\vec{s}_{last} \leftarrow \vec{s}$ |
| 6: **Return:** $\vec{s} \in \mathbb{I}^d$ | 6: **Return:** $\Delta\vec{s} \in \mathbb{I}^d$ |

In previous work (O'Connor and Welling, 2016), we used a quantization scheme which we refer to as *herding* for brevity and because of its relation to the deterministic sampling scheme in (Welling, 2009), but could otherwise be called Discrete-Time Bidirectional Sigma-Delta Modulation. The procedure is described in Algorithm 3. The input is summed into a potential $\phi$ over time until crossing a quantization threshold (in this case the $\pm\frac{1}{2}$ at which the round function changes value), and then resets.

Here we prove that Algorithm 4 is equivalent to applying Algorithm 3 to the output of a temporal difference modules. i.e. $herd(\Delta_t(x_t)) = \Delta_T(round(x_t))\forall t$.

First start by observing the following equivalence:

$$b = round(a) \Leftrightarrow |a - b| < \frac{1}{2} : b \in \mathbb{I} \tag{11}$$

We can apply this to the update rule in Algorithm 3:

$$\begin{aligned} s_t &= round(\phi_{t-1} + x_t) \in \mathbb{I} \\ \phi_t &= (\phi_{t-1} + x_t) - s_t \end{aligned} \tag{12}$$

$$\Rightarrow |\phi_t| < \frac{1}{2} \tag{13}$$

Now, if we unroll the two Equations 12 over time, with initial condition $\phi_0 = 0$, we see that.

$$\phi_t = \sum_{\tau=1}^{t} x_\tau - \sum_{\tau=1}^{t} s_\tau : \phi_t \in \mathbb{R}, x_\tau \in \mathbb{R}, s_\tau \in \mathbb{I} \tag{14}$$

Using Equations 13 and 11, respectively, we can say:

$$|\phi| = \left| \sum_{\tau=1}^{t} x_t - \sum_{\tau=1}^{t} s_\tau \right| < \frac{1}{2} \tag{15}$$

$$\Rightarrow \sum_{\tau=1}^{t} s_\tau = round\left( \sum_{\tau=1}^{t} x_\tau \right) \tag{16}$$

Which can be rearranged to solve for $s_t$.

$$s_t = round\left( \sum_{\tau=0}^{t} x_\tau \right) - round\left( \sum_{\tau=0}^{t-1} x_\tau \right) \tag{17}$$

Now if we receive inputs from a $\Delta_T$ unit: $x_\tau = u_\tau - u_{\tau-1}$ with initial condition $u_0 = 0$, then:

$$\vec{s}_t = round\left( \sum_{\tau=0}^{t} (u_\tau - u_{\tau-1}) \right) - round\left( \sum_{\tau=0}^{t-1} (u_\tau - u_{\tau-1}) \right) \tag{18}$$

$$= round\left( u_t \right) - round\left( u_{t-1} \right) \tag{19}$$

$$= \Delta_T(round(u_t)) \tag{20}$$

Leaving us with the Delta-Herding algorithm (Algorithm 4).

Therefore, if we have a linear function $w(x)$, and make use of Equation 1, then we can see that the following is true:

$$\Sigma_T(w(herd(\Delta_T(x)))) = \Sigma_T(w(\Delta_T(round(x)))) = w(\Sigma_T(\Delta_T(round(x)))) = w(round(x)) \tag{21}$$

## B  CALCULATING FLOPS

When computing the number of operations required for a forward pass, we only account for the matrix-products/convolutions (which form the bulk of computation in done by a neural network), and not hidden layer activations.

We compute the number of operations required for a forward pass of a fully connected network as follows:

For the non-discretized network, the number of flops for a single forward pass of a single data point through the network, the flop count is:

$$nFlops_{dense} = \sum_{l=0}^{L-1} (d_l \cdot d_{l+1} + (d_l - 1) \cdot d_{l+1} + d_{l+1}) = 2 \sum_{l=0}^{L-1} d_l \cdot d_{l+1} \tag{22}$$

Where $d_l$ is the dimensionality of layer $l$ (with $l = 0$ indicating the input layer). The first term counts the number of multiplications, the second the number of additions for reducing the dot-product, and the third the addition of the bias.

It can be argued that this is an unfair way to count the number of computations done by the non-discretized network because of the sparsity of the input layer (due to the zero-background of datasets like MNIST) and the hidden layers (due to ReLU units). Thus we also compute the number of operations for the non-discretized network when factoring in sparsity. The equation is:

$$nFlops_{sparse} = \sum_{l=0}^{L-1} \left( \sum_{i=0}^{N_l} ([a_l]_i \neq 0) \cdot d_{l+1} + \left( \sum_{i=0}^{N_l} ([a_l]_i \neq 0) - 1 \right) \cdot d_{l+1} + d_{l+1} \right)$$
$$= 2 \sum_{l=0}^{L-1} \sum_{i=0}^{N_l} ([a_l]_i \neq 0) \cdot d_{l+1} \tag{23}$$

Where $a_l$ are the layer activations $N_l$ is the number of units in layer $l$ and $([a_l]_i \neq 0)$ is 1 if unit $i$ in layer $l$ has nonzero activation and 0 otherwise.

For the rounding networks, we count the total absolute value of the discrete activations.

$$nFlops_{Round} = \sum_{l=0}^{L-1} \left( \sum_{i=0}^{N_l} |[s_l]_i| \cdot d_{l+1} + d_{l+1} \right) \tag{24}$$

Where $s_l$ is the discrete activations of layer $l$. This corresponds to the number of operations that would be required for doing a dot product with the "sequential addition" method described in Section 3.2.

Finally, the Sigma-Delta network required slightly fewer flops, because the bias only need to be added once (at the beginning), so its cost is amortized.

$$nFlops_{\Sigma\Delta} = \sum_{l=0}^{L-1} \sum_{i=0}^{N_l} |[s_l]_i| \cdot d_{l+1} \tag{25}$$

## C  BAKING THE SCALES INTO THE PARAMETERS

In Section 4.1, we mention that we can "bake the scales into the parameters" for ReLU networks. Here we explain that statement.

Suppose you have a function

$$f(x) = k_2 \cdot h \left( x \cdot \frac{w}{k_1} \right)$$

If our nonlinearity $h$ is homogeneous (i.e. $k \cdot h(x) = h(k \cdot x)$), as is the case for $relu(x) = max(0, x)$, we can collapse the scales $k$ into the parameters:

$$f(x) = k_2 \cdot relu(x \cdot w/k_1 + b) \tag{26}$$
$$= relu \left( x \cdot w \cdot k_2/k_1 + k_2 \cdot b \right) \tag{27}$$

So that after training scales, for a given network, we can simply incorporate them into the parameters, as: $w' = w \cdot k_2/k_1$, and $b' = k_2 \cdot b$.

## D  TEMPORAL MNIST

The Temporal MNIST dataset is a version of MNIST that is reshuffled so that similar frames end up being nearby. We generate this by iterating through the dataset, keeping a fixed-size buffer of candidates for the next frame. On every iteration, we compare all the candidates to the current frame, and select the closest one. The place that this winning candidate occupied in the buffer is then filled by a new sample from the dataset, and the winning candidate becomes the current frame. The process is repeated until we've sorted though all frames in the dataset. Code for generating the dataset can be found at: `https://github.com/petered/sigma-delta/blob/master/sigma_delta/temporal_mnist.py`

# E   MNIST RESULTS TABLE

| Setting | Net Type | Mnist KFlops Test (ds\sp) | Class error (tr\ts) | Int32-Energy (nJ) | Temp mnist KFlops Test (ds\sp) | Class error (tr\ts) | Int32-Energy (nJ) |
|---|---|---|---|---|---|---|---|
| ========== | ========== | ========== | ========== | ========== | ========== | ========== | ========== |
| Unoptimized | Original | 397 \ 107 | 0.024 \ 2.24 | 636 \ 173 | 397 \ 107 | 0.024 \ 2.24 | 636 \ 173 |
| | Round | 44 | 2.12 \ 4.21 | 4.42 | 44 | 2.12 \ 4.21 | 4.42 |
| | ΣΔ | 53 | 2.12 \ 4.21 | 5.32 | 24 | 2.12 \ 4.21 | 2.49 |
| λ=1e-10 | Original | 397 \ 107 | 0.024 \ 2.24 | 636 \ 173 | 397 \ 107 | 0.024 \ 2.24 | 636 \ 173 |
| | Round | 209 | 0.07 \ 2.39 | 20.9 | 209 | 0.07 \ 2.39 | 20.9 |
| | ΣΔ | 245 | 0.07 \ 2.39 | 24.6 | 110 | 0.07 \ 2.39 | 11 |
| λ=3.59e-10 | Original | 397 \ 107 | 0.024 \ 2.24 | 636 \ 173 | 397 \ 107 | 0.024 \ 2.24 | 636 \ 173 |
| | Round | 206 | 0.058 \ 2.3 | 20.7 | 206 | 0.058 \ 2.3 | 20.7 |
| | ΣΔ | 243 | 0.058 \ 2.3 | 24.3 | 109 | 0.058 \ 2.3 | 11 |
| λ=1.29e-09 | Original | 397 \ 107 | 0.024 \ 2.24 | 636 \ 173 | 397 \ 107 | 0.024 \ 2.24 | 636 \ 173 |
| | Round | 178 | 0.094 \ 2.42 | 17.8 | 178 | 0.094 \ 2.42 | 17.8 |
| | ΣΔ | 207 | 0.096 \ 2.42 | 20.7 | 92 | 0.094 \ 2.42 | 9.2 |
| λ=4.64e-09 | Original | 397 \ 107 | 0.024 \ 2.24 | 636 \ 173 | 397 \ 107 | 0.024 \ 2.24 | 636 \ 173 |
| | Round | 164 | 0.084 \ 2.41 | 16.4 | 164 | 0.084 \ 2.41 | 16.4 |
| | ΣΔ | 193 | 0.082 \ 2.41 | 19.4 | 87 | 0.084 \ 2.41 | 8.75 |
| λ=1.67e-08 | Original | 397 \ 107 | 0.024 \ 2.24 | 636 \ 173 | 397 \ 107 | 0.024 \ 2.24 | 636 \ 173 |
| | Round | 122 | 0.19 \ 2.55 | 12.2 | 122 | 0.19 \ 2.55 | 12.2 |
| | ΣΔ | 144 | 0.19 \ 2.55 | 14.5 | 65 | 0.19 \ 2.55 | 6.58 |
| λ=5.99e-08 | Original | 397 \ 107 | 0.024 \ 2.24 | 636 \ 173 | 397 \ 107 | 0.024 \ 2.24 | 636 \ 173 |
| | Round | 86 | 0.476 \ 2.88 | 8.66 | 86 | 0.476 \ 2.88 | 8.66 |
| | ΣΔ | 102 | 0.478 \ 2.88 | 10.3 | 47 | 0.476 \ 2.88 | 4.71 |
| λ=2.15e-07 | Original | 397 \ 107 | 0.024 \ 2.24 | 636 \ 173 | 397 \ 107 | 0.024 \ 2.24 | 636 \ 173 |
| | Round | 72 | 1.17 \ 3.28 | 7.21 | 72 | 1.17 \ 3.28 | 7.21 |
| | ΣΔ | 87 | 1.18 \ 3.28 | 8.78 | 41 | 1.17 \ 3.28 | 4.15 |
| λ=7.74e-07 | Original | 397 \ 107 | 0.024 \ 2.24 | 636 \ 173 | 397 \ 107 | 0.024 \ 2.24 | 636 \ 173 |
| | Round | 44 | 2.32 \ 4.26 | 4.49 | 44 | 2.32 \ 4.26 | 4.49 |
| | ΣΔ | 54 | 2.32 \ 4.27 | 5.46 | 26 | 2.32 \ 4.26 | 2.61 |
| λ=2.78e-06 | Original | 397 \ 107 | 0.024 \ 2.24 | 636 \ 173 | 397 \ 107 | 0.024 \ 2.24 | 636 \ 173 |
| | Round | 34 | 5.91 \ 7.37 | 3.49 | 34 | 5.91 \ 7.37 | 3.49 |
| | ΣΔ | 45 | 5.9 \ 7.37 | 4.53 | 23 | 5.9 \ 7.37 | 2.3 |
| λ=1e-05 | Original | 397 \ 107 | 0.024 \ 2.24 | 636 \ 173 | 397 \ 107 | 0.024 \ 2.24 | 636 \ 173 |
| | Round | 24 | 14.6 \ 14.6 | 2.5 | 24 | 14.6 \ 14.6 | 2.5 |
| | ΣΔ | 35 | 14.6 \ 14.6 | 3.58 | 19 | 14.6 \ 14.6 | 1.98 |

Table 1: Results on the MNIST and Temporal-MNIST datasets. MFlops indicates the number of operations done by each network. For the Original Network, the number of Flops is considered when using both (Dense / Sparse) matrix operations. The "Class Error" column shows the classification error on the training / test set respectively. The "Energy" is an estimate of the average energy that would be used by arithmetic operations per sample, if the network were implemented with all integer values. This is based on the estimates of Horowitz (2014). Again, for the Original Network, the figure is based on the numbers for dense/sparse matrix operations.

# F   HIGH-LEVEL FEATURE STABILITY

We had initially expected that, when a convolutional network is tasked with processing subsequent frames of video, high-level features would change much more slowly than the pixels and low-level features. This would give a computational advantage to our Sigma-Delta networks, whose computational cost scales with the amount of change in the feature representations. To our surprise, this appeared not to be the case. See the final plot of Figure 5. To verify that this was a property of the original convolutional network (and not somehow related our discretization scheme), we take the same snippet of video used for Figure 5 and measure the inter-frame differences. Figure 6 shows the results of this small experiment, and confirms that our initial belief - that inter-frame differences should become smaller and smaller at higher layers, was not quite correct.

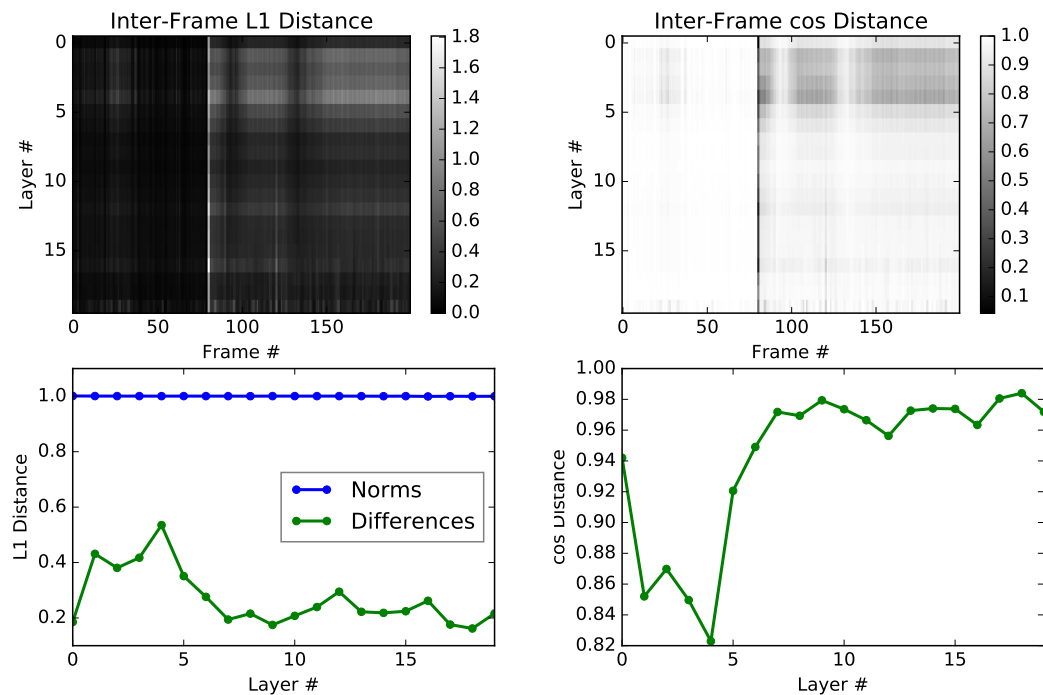

Figure 6: **Top-Left**: A heatmap showing the L1-distances between the feature representations (post-nonlinearity) of adjacent frames from the video in Figure 5 at different layers (rows) and frames (columns). The input is considered to be layer 0. Feature representations have been L1-normalized per-layer **Bottom Left**: The L1-Norms (which are 1 due to the normalization) and inter-frame L1-Distances for each layer, averaged over frames. **Top and Bottom Right**: The same measurements, with the cosine-similarity metric instead of L1. We note from these plots that the inter-frame difference is not much smaller in higher layers than it is at the pixel level, and that in the lower layers, feature representations of neighbouring frames are significantly more *dissimilar* than they are at the pixel level.

