# Peer review of "Sigma Delta Quantized Networks"

_ICLR 2017 — accepted_

[Official Review · AnonReviewer2 · rating 6 · confidence 3 · 16 Dec 2016]

The paper presents a method to improve the efficiency of CNNs that encode sequential inputs in a ‘slow’ fashion such that there is only a small change between the representation of adjacent steps in the sequence.
It demonstrates theoretical performance improvements for toy video data (temporal mnist) and natural movies with a powerful Deep CNN (VGG). 

The improvement is naturally limited by the ‘slowness’ of the CNN representation that is transformed into a sigma-delta network: CNNs that are specifically designed to have ‘slow’ representations will benefit most. Also, it is likely that only specialised hardware can fully harness the improved efficiency achieved by the proposed method. Thus as of now, the full potential of the method cannot be thoroughly evaluated.
However, since the processing of sequential data seems to be a broad and general area of application, it is conceivable that this work will be useful in the design and application of future CNNs.

All in all, this paper introduces an interesting idea to address an important topic. It shows promising initial results, but the demonstration of the actual usefulness and relevance of the presented method relies on future work.

[Official Review · AnonReviewer3 · rating 8 · confidence 4 · 16 Dec 2016]
**No Title**
originality 1 · meaningful comparison 3

This is an interesting paper about quantized networks that work on temporal difference inputs.  The basic idea is that when a network has only to process differences then this is computational much more efficient specifically with natural video data since large parts of an image would be fairly constant so that the network only has to process the informative sections of the image (video stream). This is of course how the human visual system works, and it is hence of interest even beyond the core machine learning community. 

As an aside, there is a strong community interested in event-based vision such as the group of Tobi Delbrück, and it might be interesting to connect to this community. This might even provide a reference for your comments on page 1.

I guess the biggest novel contribution is that a rounding network can be replaced by a sigma-delta network, but that the order of discretization and summation doe make some difference in the actual processing load. I think I followed the steps and 
Most of my questions have already been answers in the pre-review period. My only question remaining is on page 3, “It should be noted that when we refer to “temporal differences”, we refer not to the change in the signal over time, but in the change between two inputs presented sequentially. The output of our network only depends on the value and order of inputs, not on the temporal spacing between them.”

This does not make sense to me. As I understand you just take the difference between two frames regardless if you call this temporal or not it is a change in one frame. So this statement rather confuses me and maybe should be dropped unless I do miss something here, in which case some more explanation would be necessary.

Figure 1 should be made bigger.

An improvement of the paper that I could think about is a better discussion of the relevance of the findings. Yes, you do show that your sigma-delta network save some operation compared to threshold, but is this difference essential for a specific task, or does your solution has relevance for neuroscience?

[Official Review · AnonReviewer1 · rating 8 · confidence 4 · 17 Dec 2016]
**No Title**

This paper presented a method of improving the efficiency of deep networks acting on a sequence of correlated inputs, by only performing the computations required to capture changes between adjacent inputs. The paper was clearly written, the approach is clever, and it's neat to see a practical algorithm driven by what is essentially a spiking network. The benefits of this approach are still more theoretical than practical -- it seems unlikely to be worthwhile to do this on current hardware. I strongly suspect that if deep networks were trained with an appropriate sparse slowness penalty, the reduction in computation would be much larger.

[Public Comment · Peter OConnor · 24 Jan 2017]
**Updated Version**

Dear Reviewers.

Thank you for all your comments.  We have uploaded a final draft based on your feedback.  

Summary of changes:
- Added to the discussion section, mentioning applicability for hardware, and how training on "slow features" may improve our network, as mentioned by Reviewers 1 and 2.  
- Added new plots to the final figure, showing how computation breaks down across layers in different versions of the network. 
- Added a small experiment, in appendix, demonstrating that in the pretrained VGGnet we used, features to not change significantly more slowly in higher layers than in lower layers.
- Moved connection to herding to appendix.
- Clarified the point about Temporal vs Sequential differences brought up by Reviewer 3.

[Final Decision · Program Chairs · 06 Feb 2017]
**ICLR committee final decision**

The reviewers are in consensus that this paper introduces an interesting idea with potentially huge gains in the efficiency of video analysis tasks (dependent on hardware advances). There was extensive discussion during the question/review period. Two of the reviewers scored this paper in the Top 50% of accepted papers. The lower score is from the less confident reviewer. This seems to the AC to be a clear accept.